# Synthesis and Properties of the Gallium-Containing Ruddlesden-Popper Oxides with High-Entropy B-Site Arrangement

**DOI:** 10.3390/ma15186500

**Published:** 2022-09-19

**Authors:** Juliusz Dąbrowa, Jan Adamczyk, Anna Stępień, Marek Zajusz, Karolina Bar, Katarzyna Berent, Konrad Świerczek

**Affiliations:** 1Faculty of Materials Science and Ceramics, AGH University of Science and Technology, al. Mickiewicza 30, 30-059 Krakow, Poland; 2Faculty of Energy and Fuels, AGH University of Science and Technology, al. Mickiewicza 30, 30-059 Krakow, Poland; 3AGH Centre of Energy, AGH University of Science and Technology, ul. Czarnowiejska 36, 30-054 Krakow, Poland; 4Academic Centre for Materials and Nanotechnology, AGH University of Science and Technology, al. Mickiewicza 30, 30-059 Krakow, Poland

**Keywords:** high-entropy oxides, Ruddlesden–Popper phases, cathodes, membranes

## Abstract

For the first time, the possibility of obtaining B-site disordered, Ruddlesden–Popper type, high-entropy oxides has been proven, using as an example the LnSr(Co,Fe,Ga,Mn,Ni)O_4_ series (Ln = La, Pr, Nd, Sm, or Gd). The materials were synthesized using the Pechini method, followed by sintering at a temperature of 1200 °C. The XRD analysis indicated the single-phase, *I*4/*mmm* structure of the Pr-, Nd-, and Sm-based materials, with a minor content of secondary phase precipitates in La- and Gd-based materials. The SEM + EDX analysis confirms the homogeneity of the studied samples. Based on the oxygen non-stoichiometry measurements, the general formula of LnSr(Co,Fe,Ga,Mn,Ni)O_4+*δ*_, is established, with the content of oxygen interstitials being surprisingly similar across the series. The temperature dependence of the total conductivity is similar for all materials, with the highest conductivity value of 4.28 S/cm being reported for the Sm-based composition. The thermal expansion coefficient is, again, almost identical across the series, with the values varying between 14.6 and 15.2 × 10^−6^ K^−1^. The temperature stability of the selected materials is verified using the in situ high-temperature XRD. The results indicate a smaller impact of the lanthanide cation type on the properties than has typically been reported for conventional Ruddlesden–Popper type oxides, which may result from the high-entropy arrangement of the B-site cations.

## 1. Introduction

The perovskite-type oxides are widely considered to be one of the most important groups of functional materials, their application covering a wide range of technologies, including magnetic materials [1], catalysis [2], and, most importantly, the broadly understood energy conversion technologies [2,3,4]. The popularity of these structures stems from their unparalleled versatility, which, in turn, is a direct result of their atomic-level structure [2]. Among the different types of perovskite-related structures, the Ruddlesden–Popper type oxides (RP) have recently been garnering growing interest [3,5,6,7]. The Ruddlesden–Popper structure can be described using the general formula of (AO)(ABO_3_)*_n_* (alternatively, AO + *n*ABO_3_), in which slabs of *n*-consecutive ABO_3_ perovskite planes are alternately stacked with rock-salt-structured AO layers along the [001] direction [8]. In comparison to conventional perovskites, the RP-type phases do not preserve the three-dimensional network of corner-sharing BO_6_ octahedral units. Instead, they exhibit a two-dimensional, layered character that imposes strong anisotropy in the structure, influencing, among others, the transport and catalytical properties of these materials [3,7,9]. As a result of the above-mentioned structural factors, while the Ruddlesden–Popper phase shares multiple similarities with conventional perovskites (e.g., the strong influence of the BO_6_ units on the properties of the materials), they are also characterized by the presence of multiple, highly unique features, such as moderate thermal expansion behavior or the possibility of the formation of interstitial oxygen defects, enabling additional mechanisms of ionic transport [3,7]. Additionally, the selection of the A-site ions seems to alter their properties to a much higher degree than in the conventional perovskites, which can be seen from the example of the Ln_2_CuO_4_ series (Ln = La, Nd, Pr), where the character of conductivity changes from semiconducting to metallic-like behavior, wherein Pr is replaced by La [10]. Consequently, much effort has been invested in exploiting these features, in order to address some of the issues troubling the more popular, perovskite-structured materials.

One area in which this possibility is especially extensively researched is solid-oxide fuel cells (SOFCs), including the variation of proton ceramic fuel cells (PCFCs). Over the last few decades, SOFCs have established themselves as one of the most promising solutions for energy production, thanks to their unparallel performance with regard to energy conversion, including both chemical to electrical (fuel cell) and electrical to chemical (electrolysis) modes. This versatility makes them an especially attractive component of the renewable sources-based power grid, enabling the support of its balance, which, in most cases, is highly dependent on environmental factors. Combined with unrivaled efficiency (even up to 85% in combined heat and power systems (SHPs) [11]), negligible pollution level (when fueled by pure hydrogen), and the possibility of utilizing the existing gas infrastructure, it makes the further development of SOFC devices one of the most tempting prospects from the viewpoint of the constantly growing energy demand and the ongoing climate crisis [12]. However, as of today, the widespread popularization of this technology is still hindered by a number of unresolved issues, such as high material and manufacturing costs, insufficient longevity, and limited mobility, which all originate from the excessively high operating temperature of the state-of-the-art fuel cells [3,12]. Among the most prominent limiting factors standing in the way of widespread adoption is the insufficient performance of cathode materials [3,12], currently based almost exclusively on simple or double perovskite-type compounds [3]. The main problem here is the well-defined trade-off between the electrochemical performance and functionality of these materials, with most of the high-performing compositions being troubled by excessive thermal expansion, chemical instability, and reactivity toward not only other elements of the cell, but also gaseous species, such as CO_2_, SO_2_, and volatile Cr species, originating from the interconnected elements [13,14]. These deficiencies are the main reason why Ruddlesden–Popper phases, with their distinct, unique features, have been steadily gaining scientific attention.

It should be noted here that the strengths of RP-type materials in the role of cathodes do not lie in their electrochemical performance alone, since today, although this performance is adequate, they are still considered inferior to conventional and double perovskites [3,7,9]. To a large degree, this stems from the presence of catalytically less active AO planes, impacting the properties of the terminating surfaces of the structure [15]. On the other hand, the Ruddlesden–Popper-type oxides are characterized by much lower thermal expansion (especially for *n* = 1 compounds [7]), much better stability against, e.g., CO_2_ [3], as well as the fact that they offer the possibility of obtaining Co-free materials, potentially decreasing both the materials’ cost and their negative environmental impact (due to cobalt’s toxicity). Furthermore, while their performance in conventional SOFCs might not be spectacular, the latest studies indicate that they might excel in PCFC-type devices, offering highly competitive values of cathodic polarization resistance being reported for several compounds [7,16].

Another area in which the Ruddlesden–Popper-type oxides are becoming increasingly widespread is ceramic membranes [17], utilized, e.g., in gas purification, gas separation and recovery, auto-thermal hydrocarbon reforming, and the CO_2_-capturing process [18,19]. In this context, the mixed electronic-ionic conductivity (MIEC) of RP oxides, combined with their high-temperature stability and high CO_2_ tolerance, makes them an especially attractive option for oxygen permeation membranes. Nevertheless, similar to SOFC applications, their main advantages come from their functionality rather than their pure electrochemical performance. As a result, new ways of modifying the properties of RP oxides are constantly being pursued.

One of the solutions with particularly high potential in this regard is the application of a high-entropy approach to their design. The idea of utilizing several elements in equimolar or near-equimolar ratios appears to be especially appealing in oxide systems, as the possibility of obtaining new, unexpected properties resulting from the synergies between different elements might give access to previously unavailable properties or their combinations, potentially addressing the current limitations of state-of-the-art, functional oxide materials. Since the beginning of their development back in 2015 [20], the variety of known structures of the high-entropy oxides (HEOx) has been constantly growing, including rock-salt-structured materials [20,21], spinels [22], fluorite- and bixbyite-structured systems [23,24,25], perovskites [26,27], pyrochlores [28], magnetoplumbites [29], high-entropy garnets [30], and, most recently, high-entropy Ruddlesden–Popper oxides [31]. The range of their possible applications has been growing even more rapidly, especially in the context of energy conversion technologies, where they have already proved their potential in Li-ion and Na-ion batteries [32,33], as well as in SOFC and PCFC fuel cells [34,35,36].

In the case of the high-entropy Ruddlesden–Popper-type oxides, the first reports on the subject date back to 2020 [31], with most of the studies concerning the Ln_2_CuO_4_ cuprates, characterized by the high-entropy arrangement of the lanthanide A-site ions [31,37,38,39]. However, while, in general, the RP oxides exhibit a particularly strong correlation between a type of A-site ion and its properties [9,10,40], it can be argued that in order to fully exploit the potential benefits of the high-entropy approach in these structures, focusing on the B-site occupancy might be worthwhile since, similar to the perovskites, it is the BO_6_ units that, to a large degree, impose their transport and catalytic properties [2]. Nevertheless, to the best of the authors’ knowledge, no attempts regarding the possibility of obtaining Ruddlesden–Popper type oxides with high-entropy B-site configuration have ever been published.

In the presented study, we report for the first time the successful synthesis of the high-entropy Ruddlesden–Popper oxides (*n* = 1) with a B-site, high-entropy configuration, using as an example the LnSr(Co,Fe,Ga,Mn,Ni)O_4_ series (Ln = La, Pr, Nd, Sm, or Gd). The selection of the Co, Fe, Mn, and Ni, B-site ions was based on the well-established capability of these elements to be incorporated into RP-type structures [9]. The addition of Ga was motivated on the one hand by its constant, trivalent oxidation state, potentially influencing the tendencies of other, multivalent cations, and on the other hand, by the possibility of illustrating the potential of the high-entropy approach for incorporating within the solid-solution structure significant amounts of elements that may not appear to be optimal, based on the guidelines developed for more conventional Ruddlesden–Popper oxides. The structure and properties of the resulting materials are investigated, and the impact of the A-site lanthanide ion selection is discussed, showing a number of distinct features that likely result from the application of the high-entropy approach.

## 2. Materials and Methods

The base powders of the LnSr(Co,Fe,Ga,Mn,Ni)O_4_ (Ln = La, Pr, Nd, Sm, or Gd) oxides were synthesized using modified Pechini sol-gel methods. All considered cations were introduced in the form of highly solvable nitrate salts: La(NO_3_)_3_·6H_2_O (Alfa Aesar (Kandel, Germany), 99.9%), Pr(NO_3_)_3_·6H_2_O (Alfa Aesar, 99.9%), Nd(NO_3_)_3_·6H_2_O (Alfa Aesar, 99.9%), Sm(NO_3_)_3_·6H_2_O (Alfa Aesar, 99.99%), Gd(NO_3_)_3_·6H_2_O (Alfa Aesar, 99.99%), Sr(NO_3_)_2_ (Alfa Aesar 99.9965%), Co(NO_3_)_2_·6H_2_O (Alfa Aesar, 98–102%), Ga(NO_3_)_2_·xH_2_O (Alfa Aesar, 99.99%), Fe(NO_3_)_3_·9H_2_O (Alfa Aesar, ≥98%), Mn(NO_3_)_2_·4H_2_O (Alfa Aesar, 98%), and Ni(NO_3_)_2_·6H_2_O (Alfa Aesar, 99.9985%). In the case of gallium nitrate, the hydration level was established using thermogravimetric analysis (x = 7.88). All nitrates were dissolved in around 250 mL of distilled water. Citric acid monohydrate (Alfa Aesar 99.5+%) was further added to enable the formation of chelates, followed by ethylene glycol (EG) addition, which is necessary for the esterification reaction. The utilized molar ratios were as follows—1:2:4, for a total sum of cations, citric acid, and EG, respectively. The prepared mixtures were placed on a heated magnetic stirrer, set at 150 °C. After obtaining a clear solution, the temperature was increased to 300 °C and was kept at this level until all the water had evaporated and formed gel (around 3 h). The final product was then calcined at 700 °C for 6 h. The obtained powders were then used for the preparation of the pellets, using a uniaxial pressing method assisted by a vacuum pump. Pellets of 10 mm diameter were formed under a repeatedly applied pressure of 0.5, 1, and 1.5 tonnes. The green samples were sintered at 1200 °C for 20 h, followed by slow cooling at the rate of 2 °C/min. The density of the sintered samples was verified using the Archimedes method (with ethanol).

The X-ray diffraction (XRD) measurements were performed at room temperature (RT) using a Panalytical (Malvern, UK) Empyrean diffractometer with Cu Kα radiation, working within the Bragg-Brentano geometry, within the 10–110-degree range. The obtained data were then analyzed with the Panalytical X’Pert HighScore Plus 3.0 software, using the ICDD PDF2 database. The same software was used for all the Rietveld refinements. For selected materials, additional, in situ, high-temperature (HT-XRD) measurements were carried out, using the Anton Paar HTK 1200N furnace mounted on the diffractometer. The obtained results enabled assessing the high-temperature stability of the studied compositions, as well as providing additional information regarding their thermomechanical behavior.

The chemical composition, homogeneity, and microstructure were examined using scanning electron microscopy (SEM), combined with energy-dispersive X-ray spectroscopy (EDX). Then, point, area, and elemental map EDX analyses were performed. Most measurements were carried out using the ThermoFisher Scientific Phenom XL Desktop SEM (Waltham, MA, USA), equipped with a silicon drift detector and ProSuite analytic software, also supplied by ThermoFisher (Waltham, MA, USA). The analysis, especially in terms of the average chemical composition, was further supplemented by the data collected using the FEI (Hillsboro, OR, USA) Versa 3D microscope, equipped with the more precise Oxford Instruments (Abingdon, UK) Ultim Max EDX spectroscope. For all analyses, the accelerating voltage was equal to 15 kV.

To provide further information regarding the morphology of the sinters, atomic force microscopy (AFM) measurements were performed for single-phase materials. The measurements were carried out using the Bruker (Billerica, MA, USA) MultiMode VIII microscope with an antimony-doped silicon probe (nominal tip radius: 8 nm, spring constant: 40 N/m). Tapping mode (semi-contact mode) was utilized.

The room-temperature oxygen non-stoichiometry within the studied series of compositions was determined with the use of the iodometric titration method [41]. All measurements were performed under an Ar atmosphere, using an EM40-BNC Mettler Toledo (Columbius, OH, USA) titrator. The experimental procedure involved several steps, including weighing approximately 0.05 g of the examined powder and 1 g of KI (in excess), mixing them together, and adding a magnetic stirrer bar to the beaker. Then, while stirring, the addition of 5 mL of demineralized water and 8 mL of 6 M HCl, followed by another 100 mL of demineralized water, was performed. The last step involved titration with Na_2_S_2_O_3_ until the color of the solution changed from yellow to a colorless fluid, indicating that it had reached the endpoint of the titration, which was precisely determined using the potentiometric method. The occurring reactions can be summarized as follows:$$\begin{array}{c} Me^{y+}+\left(y-2\right)I^{-}\to Me^{2+}+\frac{y-2}{2}I_{2}\\ I_{2}+2S_{2}O_{3}^{2-}\to 2I^{-}+S_{4}O_{6}^{2-} \end{array}$$
where: Me—multivalent cation (Co/Fe/Mn/Ni), and y—average oxidation state. All calculations were conducted assuming a constant 3+ valence state of Ga. For each material, three separate measurements were carried out to provide necessary statistics. To further improve the reliability of the data, all powders were annealed at 300 °C overnight, directly before the measurement occurred, to minimize the errors associated with the possible hydration.

The temperature dependence of the oxygen non-stoichiometry was assessed for the single-phase materials via thermogravimetric analysis (TA Instruments (New Castle, DE, USA) Q5000 IR thermobalance). For each sample, two consecutive heating and cooling runs were measured, to ensure the equilibrium nature of the data (the data from the second cooling run was considered representative). The measurements were performed within the RT—800 °C temperature range, with 10 °C min^−1^ heating and cooling rates. During the measurement process, synthetic air was supplied at the rate of 100 cm^3^/min. 

The temperature dependence of the total electrical conductivity was studied by the 4-probe DC method within a temperature range of 25–900 °C. The utilized Fine Instruments (Kraków, Poland) FRASB-1000 setup consists of: TF1200 tube furnace, Keysight Multimeter 34465A, Keysight Function/Arbitrary Waveform Generator 33210A, control unit CFRASB-1000 with AM16/32B multiplexer, measuring probe SSC-15, and BR07 retaining decade. The measurements were performed on cuboid samples that were covered at the parallel ends with platinum paste to improve the electrical contact. Additionally, the Fine Instruments FRASB-1000 apparatus allows for measuring the Seebeck coefficient values, providing further information regarding the nature of charge carriers.

The thermomechanical behavior of the materials was determined using the Linseis (Robbinsville, NJ, USA) L75 Platinum Series dilatometer. The samples with cuboid shapes were positioned between alundum discs and then pressed with a force of 300 mN. The measurements were performed for both heating and cooling runs in the temperature range from 25 to 900 °C, with a ramp rate of 5 °C/min. The obtained data were further normalized using the correction curve, to compensate for the thermal expansion of the alundum discs.

## 3. Results and Discussion

The XRD diffractograms of the sintered samples, together with the results of Rietveld refinement and the procedure’s residuals, are presented in Figure 1 and Table 1. The theoretical and experimental densities are also provided, with the exception of the La-based material, in which case, the amount of secondary phase was high enough to impact the reliability of the measurement.

As can be seen in all cases, the main phase is characterized by the aristotype *I*4/*mmm* symmetry, with characteristics of the *n* = 1, Ruddlesden–Popper, *T*-type phase. This is further supported by the relatively high values of the Goldschmidt parameter, *t*, varying from 0.936 to 0.955 across the series (for Gd- and La-based compositions, respectively), which are usually correlated with the formation of a *T*-type structure over *T*’ one [8]. Out of 5 studied materials, three can be considered phase-pure, namely, PrSr(Co,Fe,Ga,Mn,Ni)O_4_, NdSr(Co,Fe,Ga,Mn,Ni)O_4_, and SmSr(Co,Fe,Ga,Mn,Ni)O_4_, with LaSr(Co,Fe,Ga,Mn,Ni)O_4_ and GdSr(Co,Fe,Ga,Mn,Ni)O_4_, based on the largest and the smallest considered lanthanide cation, respectively, containing a minor amount of secondary phases (5.2 and 0.4 wt%, respectively). Still, based on their small fractions, these compositions were qualified for further characterization. As can be expected, with the increasing size of the lanthanide cation, both the value of the *c*-lattice parameter (reflecting the elongation of the interlayer distances in the structure) and the unit cell volume increased, with the normalized quasi-cubic lattice parameter *a*_0_ exhibiting an almost linear dependence upon the Ln ionic radius (see Appendix A). An exception here is the La-based material, in which the unit cell size appears to be slightly bigger than expected, based on the trend within the series.

The microstructure and chemical composition of the studied materials was assessed using the SEM + EDX method. The exemplary micrograph for the NdSr(Co,Fe,Ga,Mn,Ni)O_4_ composition, together with the corresponding results of EDX mapping and point analysis, is presented in Figure 2. Data for other materials can be found in Appendix A. The corresponding results of the AFM measurements can be found in Appendix A. The average chemical compositions for all materials are summarized in Table 2. As can be seen, the B-site occupancy is close to the nominal one, although the content of Ga appears to be slightly higher across the whole series. The likely reason for that is the fact that some of the lines of the Ga EDX spectra might overlap, e.g., with the *Mα*_1_ lines of the lanthanides, decreasing the accuracy for this element, as indicated by the relatively high error values. Another contribution might be the fact that the variable hydration level of Ga(NO_3_)_3_, even though it was established pre-synthesis via TG measurements, might introduce certain inaccuracies.

The morphology of all prepared pellets indicated the presence of elongated, needle-like grains, which is typical for the Ruddlesden–Popper-type phases. In all cases, including the LaSr(Co,Fe,Ga,Mn,Ni)O_4_ and GdSr(Co,Fe,Ga,Mn,Ni)O_4_, which were assessed as multiphase systems, based on the XRD results, the EDX mappings and point analysis suggest the presence of a single-phase structure with only minor, local variations of Sr and Ga content in some of the samples. Still, all the materials exhibited generally high chemical homogeneity, with a lack of apparent signs of phase separation. The microstructure of the studied samples, especially in terms of the observed grain growth, indicates a small improvement in the sinterability of the materials with the decreasing size of the lanthanide ion, which finding is in line with the relative density values presented in Table 1.

This oxygen non-stoichiometry behavior is one of the most distinct structural features of the Ruddlesden–Popper phases. As the coordination number of the A-site cation within the AO layers decreases from 12 in perovskite-type layers to 9 [8], empty tetrahedral positions are created within the AO planes, enabling the presence and movement of oxygen interstitials. As a result, the oxygen ion conductivity in RP oxides might occur through both interstitial and vacancy-mediated mechanisms [7,42]. In most cases, the level of oxygen non-stoichiometry strongly depends not only on the size of the A-site ions and the type of B-site cations but also on the external conditions (mainly temperature and oxygen partial pressure), even leading to a change in the material’s behavior from hypo- to hyperstoichiometric [7]. It is, therefore, particularly interesting to investigate the oxygen non-stoichiometry in the presented LnSr(Co,Fe,Ga,Mn,Ni)O_4_ series since one might expect that the combination of the presence of multiple redox pairs on the B-sites, combined with the changing ionic radius of the A-site lanthanides, may facilitate a wide range of different behaviors. Consequently, the room-temperature oxygen non-stoichiometry was evaluated for all materials in the series, using the iodometric titration method, the obtained results being summarized in Figure 3a. The exact values from each series of measurements can be found in Appendix A. Additionally, for the single-phase materials, the temperature dependence of oxygen non-stoichiometry was also determined, the results from the thermogravimetric analysis being presented in Figure 3b.

As can be seen, the results indicate rather unintuitive behavior, as the level of oxygen non-stoichiometry is almost identical to the method’s error for all the studied compositions. All materials display slight oxygen hyperstoichiometry, varying from 0.006 to 0.015, for LaSr(Co,Fe,Ga,Mn,Ni)O_4_ and PrSr(Co,Fe,Ga,Mn,Ni)O_4_, respectively, leading to a general formula of LnSr(Co,Fe,Ga,Mn,Ni)O_4+*δ*_. The close-to-stoichiometric nature of the studied series is in line with the behavior reported for conventional, TM-based Ruddlesden–Popper oxides, heavily doped with Sr (*x* ≥ 1), such as La_1−*x*_Sr*_x_*MnO_4±*δ*_ [43], La_1−*x*_Sr*_x_*CoO_4±*δ*_ [44] or La_1−*x*_Sr*_x_*NiO_4±*δ*_ [44]. The observed variation in *δ* values does not suggest the presence of any trend that could be associated with the size of the A-site lanthanide cation. Such behavior is not completely atypical for RP-type oxides as, for example, in the oxygen hyperstoichiometric series of Ln_2_CuO_4+δ_ (Ln = La, Pr, and Nd), the values of δ vary between 0.03 to 0.05 [10], while in the case of Ln_2_NiO_4+δ_ (Ln = La, Pr, and Nd), δ values from 0.13 to 0.16 are reported [45]. Still, in comparison, the observed spread of δ values in the LnSr(Co,Fe,Ga,Mn,Ni)O_4+*δ*_ series can be considered particularly low. In terms of the temperature dependence of oxygen non-stoichiometry, for all single-phase materials, the thermogravimetric analysis indicates negligible changes within the RT to the 800 °C range. This is to be expected for near-stoichiometric materials. Unfortunately, little data is available in this regard for conventional RP-type materials with a high level of alkali doping, not to mention the influence of the Ln type, as most of the studies considered mainly the oxygen partial pressure dependence [43,44,46]. To sum up, the observed oxygen non-stoichiometry behavior is in line with expectations, based on the behavior of conventional RP-type oxides, although the observed spread of *δ* can still be considered relatively low by such standards. This suggests that the high-entropy configuration of the B-site ions within the studied series may possess a slightly stronger influence over the level of oxygen non-stoichiometry than the selection of the A-site rare-earth cations.

The prevailing influence of the B-site ions over the A-site ions can be also observed in the case of transport properties. The measured values of total conductivity for the LnSr(Co,Fe,Ga,Mn,Ni)O_4+*δ*_ series are presented in Figure 4. The energies of activation *E_a_* are determined using the Arrhenius equation:(1)$$\sigma_{tot}T=\sigma_{0}\exp\left(\frac{-E_{a}}{kT}\right),$$
where *k* denotes the Boltzmann constant, and are summarized in Table 3, together with the values of maximum conductivity, *σ_max_*.

It should be noted that in the case of RP-type oxides, the selection of the A-site cations may have a particularly strong impact on the transport properties, influencing not only the values of the total conductivity but also even its type [9,10,40]. However, in the case of the studied series, the temperature dependencies of the total conductivity are remarkably similar for all Ln cations, both in terms of maximum conductivity values (ranging from 2.45 to 4.28 S/cm for LaSr(Co,Fe,Ga,Mn,Ni)O_4+*δ*_ and SmSr(Co,Fe,Ga,Mn,Ni)O_4+*δ*_, respectively) and the energies of activation. In all cases, the materials display typical, semiconducting behavior, with distinguishably different energies of activation for low- and high-temperature ranges, indicating a change in the conductivity mechanism. The determined values of total conductivity are relatively low for RP oxides but are still within the expected range, being superior to, e.g., LaSrCoO_4_ or La_1.4_Sr_0.6_FeO_4_ [9]. On the other hand, the values of the activation energies are relatively high, and the character of temperature dependence is rather atypical, showing some similarities to Sm_2_CuO_4_ oxide [40]. The only material that marginally stands out among the rest of the series is the SmSr(Co,Fe,Ga,Mn,Ni)O_4+*δ*_, characterized by the highest conductivity values across the whole investigated temperature range, combined with the lowest values of activation energies.

The values of the Seebeck coefficient were also measured to provide further insight into the conductivity behavior. The results are presented in Figure 5.

As can be seen, in all cases, the value of the Seebeck coefficient is negative within the whole investigated temperature range. While this may appear surprising, considering the likely *p*-type of conductivity, which is typical in RP structures [9], it can be explained within the framework of the model proposed for the transition metals-based, RP-type oxides by Goodenough [47]. According to this model, the Fermi level in such materials is situated with the bandgap (at room and low temperatures), while each 3*d* hole per TM^2+^ ion is ordered into bonding σ states, which promotes the effective masses of the hole charge carriers being higher than the electron effective masses. In such a situation, despite the holes being the majority charge carriers, the materials will display the negative values of the Seebeck coefficient, especially within the range of low to intermediate temperatures, where the conductivity is intrinsic [47]. Such a behavior can be further supported by the very low level of oxygen hyperstoichiometry within the studied series, which indicates a relatively low concentration of holes, according to the electroneutrality condition for hyperstoichiometric RP oxides [46]. At higher temperatures, the Fermi energy, *E_f_*, moves toward the *d*-bands [47], leading to a decrease in the hole charge carriers’ effective mass, which, in turn, results in less-negative values of the Seebeck coefficient, as observed in Figure 5. However, while the presented description provides a general outline of the presented behavior, it does not explain the visible spread of the room-temperature values of Seebeck coefficients within the series. In this case, additional factors should be considered. The selection of the lanthanide ion may influence the level of distortion of the BO_6_ octahedra, affecting the bandstructure within the Fermi level, which, in turn, may lead to changes in the effective masses of both electron and hole charge carriers, and the resulting Seebeck coefficient values. Secondly, the type of lanthanide ion might also affect the formation of the hole itself, similarly as in the case of the Ln_2_CuO_4_ series, where, for La, the hole is ordered into the $\sigma_{x^{2}-y^{2}}^{*}$ states, while for the Pr, Nd, and Sm-based compounds, the holes are ordered into $d_{z^{2}}$ states [47]. Although the behavior illustrated in Figure 5 does not indicate nearly as strong an influence of the lanthanide’s selection, it is likely that the differences associated with the introduction of different Ln ions will impact the effective mass and mobility of charge carriers. An interesting observation can also be made by comparing the total conductivity values with the Seebeck coefficients. It can be noted that the sequence of materials, from the best to the poorest conductor (Sm > Nd > La > Pr > Gd), can be directly correlated with the sequence of Seebeck coefficient values at room temperature, from the least to the most negative (Sm > Nd > La > Pr > Gd). Furthermore, more extensive studies into these behaviors, including the temperature dependence of the magnetic properties, should follow to fully explain the observed phenomena. 

The thermal expansion behavior of the studied materials was investigated using dilatometric measurements. The results are presented in Figure 6, with the TEC values being summarized in Table 4.

As can be seen, very similar values are observed across the whole series, with thermal expansion coefficients ranging from 14.55 to 15.21 × 10^−6^ K^−1^ (based on the cooling runs). The determined TECs can be considered to be high for *n* = 1 RP oxides, for which values in excess of 14 × 10^−6^ K^−1^ are rarely reported [9]. The impact of the A-site cations appears to be almost negligible; however, in this case, such behavior is not uncommon in Ruddlesden–Popper-type oxides, which, in principle, show a much smaller variance of TECs than the conventional or double perovskites, in which the occupancy of the A-site often has a very strong influence on thermal expansion behavior [3,41,48,49]. It should also be noted that in the context of MIEC materials, the values of TEC reported here can still be considered moderate, which, combined with the lack of chemical expansion behavior, is a major advantage from the viewpoint of potential applications, e.g., in SOFCs.

Another important feature in this context is the thermal stability of the materials. In this case, the SmSr(Co,Fe,Ga,Mn,Ni)O_4+*δ*_ and NdSr(Co,Fe,Ga,Mn,Ni)O_4+*δ*_ compositions, characterized by relatively good electrical behavior and phase-pure structures, were selected. The results of the HT-XRD measurements are presented in Figure 7. The more detailed structural data can be found in Appendix A, while the thermal expansion behavior, estimated based on the unit cell expansion, is shown in Appendix A.

Based on the diffraction data, both materials are fully stable from RT up to 1000 °C, with no indication of secondary phase formation in either of them. The values of the thermal expansion coefficient appear to be slightly higher than in the case of dilatometric measurement, these being equal to 16.00 and 16.21 for SmSr(Co,Fe,Ga,Mn,Ni)O_4+*δ*_ and NdSr(Co,Fe,Ga,Mn,Ni)O_4+*δ*_, respectively, further supporting the notion of relatively high TEC values within the studied series, when compared with more conventional *n* = 1 Ruddlesden–Popper oxides. It is important that the observed stability makes it possible to consider these materials for high-temperature applications, either in fuel cell devices or as ceramic membranes.

## 4. Conclusions

The possibility of obtaining B-site-disordered, Ruddlesden–Popper-type, high-entropy oxides is proven for the first time, using the example of the LnSr(Co,Fe,Ga,Mn,Ni)O_4+*δ*_ (Ln = La, Pr, Nd, Sm, or Gd) series. The materials were synthesized by the modified Pechini method, followed by multi-stage thermal treatment. Based on the obtained results, the following conclusions can be drawn:All materials sintered at 1200 °C are characterized primarily by the *I*4/*mmm*, Ruddlesden–Popper (*n* = 1), *T*-type phase structure, with the secondary phases being observed only in the case of La- and Gd-based compositions (5.2 and 0.4 wt%, respectively). The expansion of the unit cell follows a linear trend across the series, with the exception of the La-based composition, for which the lattice parameters are slightly bigger than the trend in the rest of the series would suggest.The SEM + EDX data further supported the XRD results, showing a lack of secondary phase precipitates in all the studied materials, including LaSr(Co,Fe,Ga,Mn,Ni)O_4+*δ*_ and GdSr(Co,Fe,Ga,Mn,Ni)O_4+*δ*_. Based on the EDX mappings, the materials are highly homogenous, although the point analysis data indicate slight deviations in the content of Ga; however, there are no signs of phase separation. The sinterability of the materials improves with the decreasing ionic radius of the lanthanide ions, which is in agreement with the relative density data obtained by the Archimedes method.The oxygen non-stoichiometry behavior reveals a very low, nearly identical level of oxygen excess, varying from δ equaling 0.06 to 0.015 for all materials, with no correlation to the size of the A-site lanthanide ion. Furthermore, the level of oxygen content is nearly temperature-independent for all single-phase materials. The observed behavior indicates a relatively high impact of B-site occupancy on the properties of the materials, in comparison to the influence of the A-site cations.The electrical behavior also appears to be highly unique for RP-type materials, with only small differences between the different materials in terms of both maximum total conductivity value (varying from 2.44 to 4.28 S/cm for GdSr(Co,Fe,Ga,Mn,Ni)O_4+*δ*_ and SmSr(Co,Fe,Ga,Mn,Ni)O_4+*δ*_, respectively), and energies of activation. In all cases, the materials are characterized by a semiconducting behavior, with visible differences between low- and high-temperature ranges. This is contrary to the behavior reported for the conventional RP systems, in which the selection of the A-site lanthanide often has a profound impact on both the value and type of conductivity.The values of the Seebeck coefficient for all materials are negative within the whole investigated temperature range, varying from −50 to −130 μV/K. Despite the likely *p*-type character of the studied materials, such results can be explained based on the theories developed for conventional, RP-type oxides. It can be assumed that the determined, negative values of the Seebeck coefficient can be related to the effective masses of electron and hole charge carriers, which in turn, impact the sign of the Seebeck coefficient. Interestingly, a direct correlation between the total electrical conductivity and RT value of the Seebeck coefficient value can be observed, with the sequence in both cases being Sm > Nd > La > Pr > Gd.The thermal expansion behavior is, again, remarkably similar across the whole series, with the thermal expansion coefficient varying from 14.55 to 15.21·10^−6^ K^−1^. The observed values, while relatively high for the *n* = 1 RP materials, in general, show similar behavior to that observed in conventional systems, in which a relatively small influence of the lanthanide type on the TEC values is typically reported.The selected SmSr(Co,Fe,Ga,Mn,Ni)O_4+*δ*_ and NdSr(Co,Fe,Ga,Mn,Ni)O_4+*δ*_ materials, based on the performed HT-XRD measurements, are characterized by high thermal stability within the RT–1000 °C range.

The studied LnSr(Co,Fe,Ga,Mn,Ni)O_4+*δ*_ (Ln = La, Pr, Nd, Sm, or Gd) series represents the first case of the Ruddlesden–Popper-type oxides, with the high-entropy configuration of the B-site cations. Based on the obtained results, it can be stated that the application of the high-entropy approach in these materials resulted in certain, distinct features when compared with their more conventional counterparts, indicating a more profound impact on the B-site occupancy over the A-site one than is typically reported for these types of oxide systems. However, at this stage of study, it is impossible to state whether the reported features will be applicable to all high-entropy RP oxides or if they are just characteristic of the studied series of materials. Nevertheless, the presented materials already appear to possess potentially beneficial characteristics, from the viewpoint of high-temperature application, and, what is even more important, the reported unique behavior might provide new tools for design and tailoring the properties of the Ruddlesden–Popper phases.

## Figures and Tables

**Figure 1 materials-15-06500-f001:**
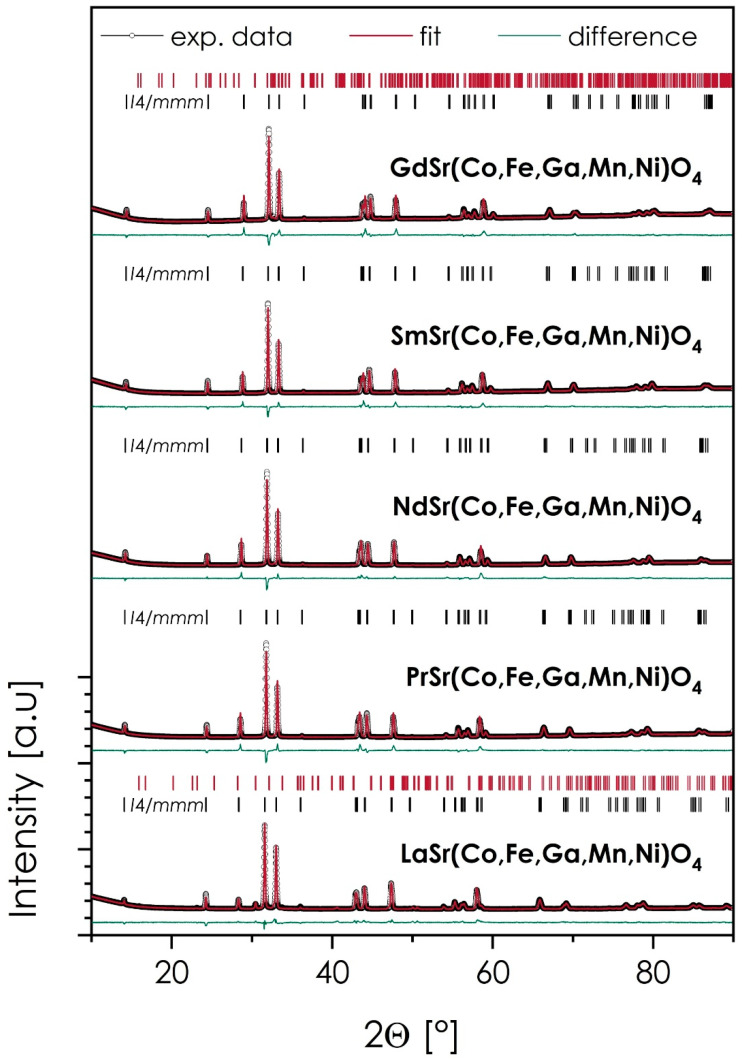
XRD diffractograms, together with the results of Rietveld refinement for the LnSr(Co,Fe,Ga,Mn,Ni)O_4_ (Ln = La, Pr, Nd, Sm, or Gd) materials, sintered at 1200 °C for 20 h.

**Figure 2 materials-15-06500-f002:**
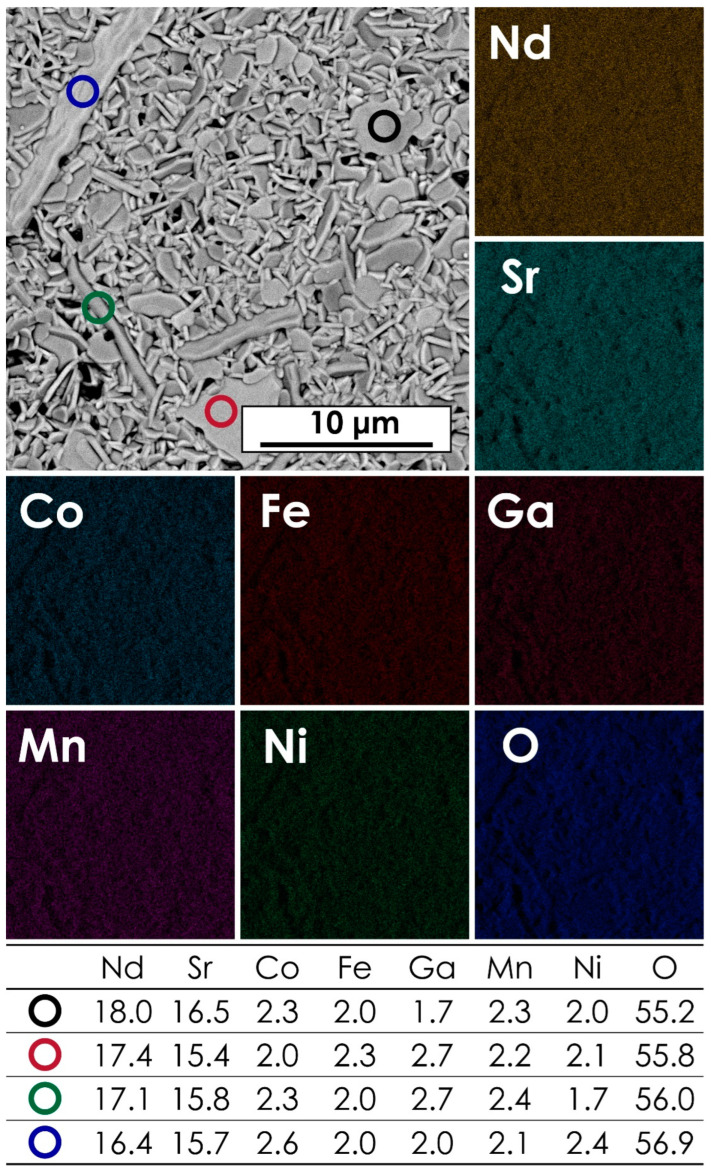
The results of EDX mapping and point analysis for the NdSr(Co,Fe,Ga,Mn,Ni)O_4_ pellet, sintered at 1200 °C for 20 h.

**Figure 3 materials-15-06500-f003:**
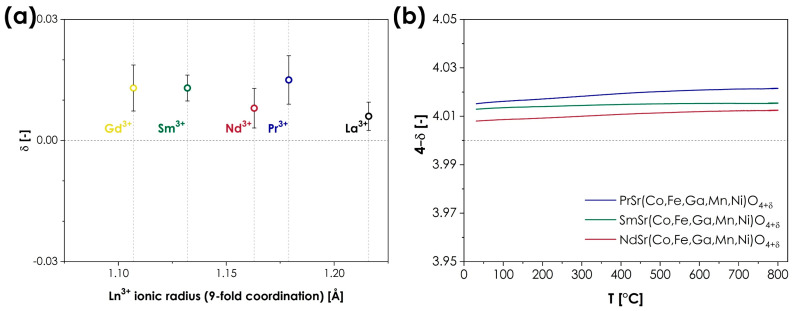
(**a**) Room temperature oxygen non-stoichiometry in the LnSr(Co,Fe,Ga,Mn,Ni)O_4+*δ*_ (Ln = La, Pr, Nd, Sm, or Gd) series, in a function of ionic radius of Ln^3+^ cations; (**b**) temperature dependence of oxygen non-stoichiometry for the single-phase compositions.

**Figure 4 materials-15-06500-f004:**
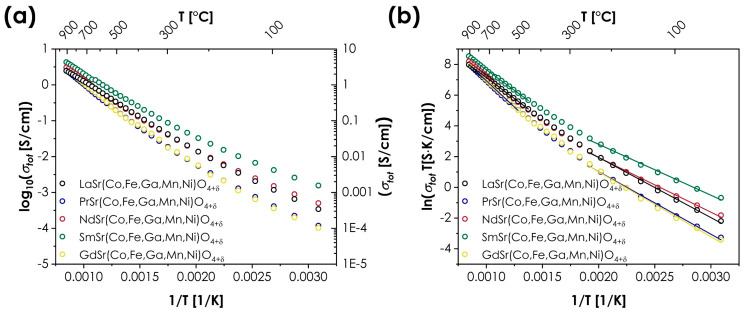
Total conductivity values for the LnSr(Co,Fe,Ga,Mn,Ni)O_4+*δ*_ series (Ln = La, Pr, Nd, Sm, or Gd): (**a**) as log_10_(*σ*) = *f*(1/*T*); (**b**) as ln(*σT*) = *f*(1/*T*).

**Figure 5 materials-15-06500-f005:**
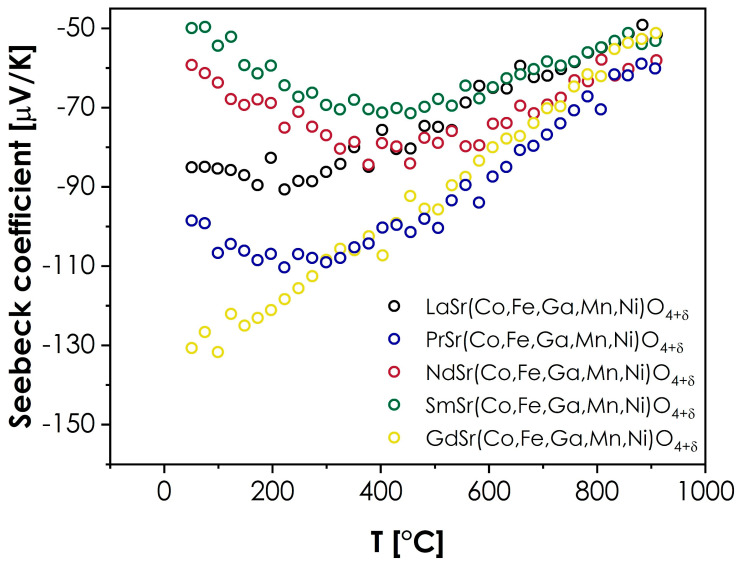
Seebeck coefficient values measured for the LnSr(Co,Fe,Ga,Mn,Ni)O_4+*δ*_ series (Ln = La, Pr, Nd, Sm, or Gd).

**Figure 6 materials-15-06500-f006:**
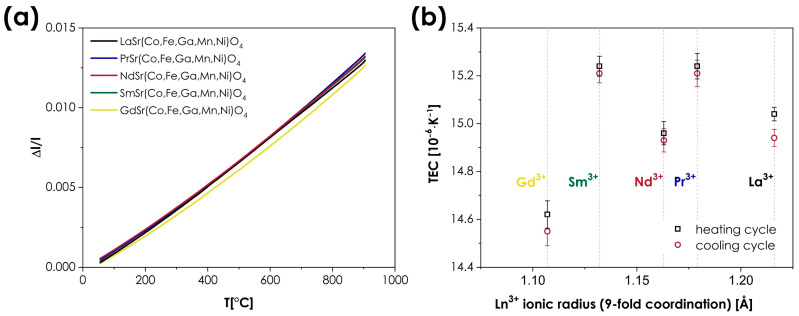
(**a**) Relative expansion of the LnSr(Co,Fe,Ga,Mn,Ni)O_4+*δ*_ (Ln = La, Pr, Nd, Sm, or Gd) materials (based on the cooling runs); (**b**) the determined TEC values, presented as a function of the Ln^3+^ ionic radii.

**Figure 7 materials-15-06500-f007:**
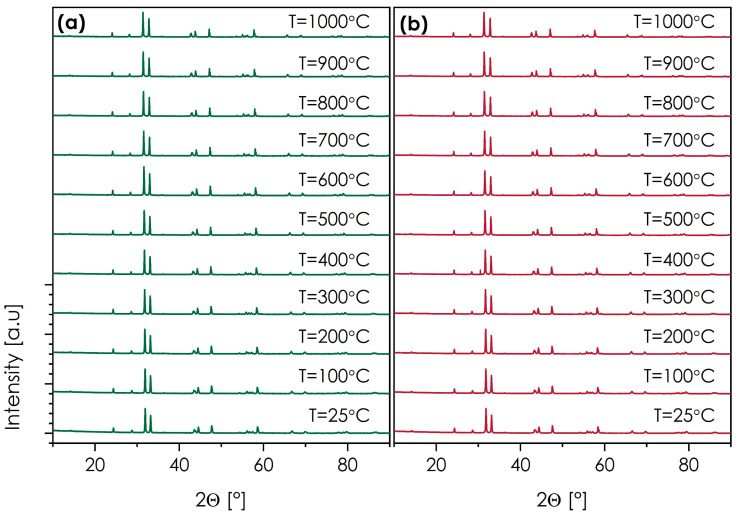
HT-XRD diffractograms (cooling run) measured for: (**a**) SmSr(Co,Fe,Ga,Mn,Ni)O_4+*δ*_; (**b**) NdSr(Co,Fe,Ga,Mn,Ni)O_4+*δ*_.

**Table 1 materials-15-06500-t001:** The results of the Rietveld refinement analysis conducted for the LnSr(Co,Fe,Ga,Mn,Ni)O_4_ (Ln = La, Pr, Nd, Sm, or Gd) materials, sintered at 1200 °C for 20 h, together with the procedure’s residuals. The density of the pellets, as determined by the Archimedes method, is also provided.

	Space Group	Content (wt%)	*a* [Å]	*b* [Å]	*c* [Å]	*V* [Å^3^]	*a*_0_ [Å]	GoF	Rwp	*δ*_the_ (g/cm^3^)	*δ*_exp_ (g/cm^3^)	*δ*_rel_ (%)
LaSr(Co,Fe,Ga,Mn,Ni)O_4_	*I*4/*mmm*	94.8	3.8393(1)	3.8393(1)	12.6075(3)	185.834(1)	5.7066(1)	5.25	4.96	6.26	-	-
	P-421 m	5.2	7.885(1)	7.885(1)	5.313(2)	-	-			-	-	-
PrSr(Co,Fe,Ga,Mn,Ni)O_4_	*I*4/*mmm*	100	3.8164(1)	3.8164(1)	12.4914(3)	181.939(1)	5.6664(1)	10.78	4.77	6.43	5.76	89.6
NdSr(Co,Fe,Ga,Mn,Ni)O_4_	*I*4/*mmm*	100	3.8101(1)	3.8101(1)	12.4470(3)	180.692(1)	5.6534(1)	12.09	4.93	6.53	5.75	88.1
SmSr(Co,Fe,Ga,Mn,Ni)O_4_	*I*4/*mmm*	100	3.8003(1)	3.8003(1)	12.3757(3)	178.734(1)	5.6329(1)	7.51	3.66	6.72	6.30	93.8
GdSr(Co,Fe,Ga,Mn,Ni)O_4_	*I*4/*mmm*	99.6	3.7946(1)	3.7946(1)	12.3157(4)	177.337(1)	5.6182(1)	7.50	3.27	6.90	6.52	94.6
	Iba	0.4	11.214(1)	18.883(1)	5.527(1)	-	-			-		

**Table 2 materials-15-06500-t002:** The chemical composition of the studied LnSr(Co,Fe,Ga,Mn,Ni)O_4_ (Ln = La, Pr, Nd, Sm, or Gd) series was determined by EDX area analysis. For each sample, three different areas were measured.

	Ln	Sr	Co	Fe	Ga	Mn	Ni	O (at. %)
LaSr(Co,Fe,Ga,Mn,Ni)O_4_	17.6(1)	14.1(1)	3.5(4)	3.5(2)	4.2(2)	3.3(2)	3.5(1)	50.4(3)
PrSr(Co,Fe,Ga,Mn,Ni)O_4_	17.0(1)	14.4(1)	3.3(1)	3.1(1)	4.2(5)	3.4(1)	3.4(3)	51.2(2)
NdSr(Co,Fe,Ga,Mn,Ni)O_4_	17.1(2)	14.3(1)	3.2(1)	3.2(2)	4.2(4)	3.4(1)	3.4(2)	51.4(2)
SmSr(Co,Fe,Ga,Mn,Ni)O_4_	17.2(2)	14.3(1)	3.3(3)	3.4(1)	3.9(4)	3.2(1)	3.2(1)	51.7(4)
GdSr(Co,Fe,Ga,Mn,Ni)O_4_	16.6(2)	14.1(1)	2.9(2)	3.4(1)	4.0(5)	3.2(1)	3.2(1)	52.6(3)

**Table 3 materials-15-06500-t003:** The determined values of energy of activation for the LnSr(Co,Fe,Ga,Mn,Ni)O_4+*δ*_ series (Ln = La, Pr, Nd, Sm, or Gd). The maximum value of total conductivity is also provided.

	*E_a_* (eV)	T (°C)	*σ_max_* (S/cm)
LaSr(Co,Fe,Ga,Mn,Ni)O_4+_*_δ_*	0.34(1)	50–250	2.45
	0.47(1)	450–900	
PrSr(Co,Fe,Ga,Mn,Ni)O_4+_*_δ_*	0.35(1)	50–250	2.54
	0.59(1)	500–900	
NdSr(Co,Fe,Ga,Mn,Ni)O_4+_*_δ_*	0.31(1)	50–250	3.05
	0.53(1)	575–900	
SmSr(Co,Fe,Ga,Mn,Ni)O_4+_*_δ_*	0.29(1)	50–250	4.28
	0.47(1)	450–900	
GdSr(Co,Fe,Ga,Mn,Ni)O_4+_*_δ_*	0.36(1)	50–250	2.74
	0.60(1)	625–900	

**Table 4 materials-15-06500-t004:** Thermal expansion coefficient values, determined by dilatometric studies for LnSr(Co,Fe,Ga,Mn,Ni)O_4+*δ*_ (Ln = La, Pr, Nd, Sm, or Gd) series.

	TEC (10^−6^·K^−1^)	
	Heating	Cooling
LaSr(Co,Fe,Ga,Mn,Ni)O_4+*δ*_	15.04 ± 0.03	14.94 ± 0.04
PrSr(Co,Fe,Ga,Mn,Ni)O_4+*δ*_	15.24 ± 0.05	15.21 ± 0.06
NdSr(Co,Fe,Ga,Mn,Ni)O_4+*δ*_	14.96 ± 0.05	14.94 ± 0.05
SmSr(Co,Fe,Ga,Mn,Ni)O_4+*δ*_	15.24 ± 0.04	15.21 ± 0.04
GdSr(Co,Fe,Ga,Mn,Ni)O_4+*δ*_	14.62 ± 0.06	14.55 ± 0.06

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
