# Peer review of "Synthesis and Properties of the Gallium-Containing Ruddlesden-Popper Oxides with High-Entropy B-Site Arrangement"

_materials, 2022, doi:10.3390/ma15186500_

Round 1
Reviewer 1 Report
In this work, the authors systemically investigated the structures of LnSr(Co,Fe,Ga,Mn,Ni)O4 (Ln = La, Pr. Nd, Sm, or Gd) by XRD, SEM, EDS, conductivity, thermal expansion coefficient. The manuscript is well written and well supported by various characterization results. I recommend its acceptance after minor revision.
1) The authors mentioned that the room-temperature oxygen non-stoichiometry was evaluated for all materials in the series using the iodometric titration method, the authors should provide more information about the method and provide related references.
2) How was the oxygen non-stoichiometry changed with the increasing temperature or converting atmosphere? Will the change in oxygen be related with the conductivity values at different temperatures?
Author Response
Dear Reviewer 1, thank you for your time and effort dedicated to our manuscript. Below, are presented our answers to the issues raised in the review.
In this work, the authors systemically investigated the structures of LnSr(Co,Fe,Ga,Mn,Ni)O4 (Ln = La, Pr. Nd, Sm, or Gd) by XRD, SEM, EDS, conductivity, thermal expansion coefficient. The manuscript is well written and well supported by various characterization results. I recommend its acceptance after minor revision.
- The authors mentioned that the room-temperature oxygen non-stoichiometry was evaluated for all materials in the series using the iodometric titration method, the authors should provide more information about the method and provide related references.
We would like to thank the Reviewer for this remark, as in the process, we found a calculation error in the titration data, which was corrected by introducing the assumption that Ga3+ is not reduced during the experimental procedure. Consequently, the values of oxygen non-stoichiometry and associated discussion were updated. Also, further information concerning the method itself was added to the methodology section.
- How was the oxygen non-stoichiometry changed with the increasing temperature or converting atmosphere? Will the change in oxygen be related with the conductivity values at different temperatures?
The data concerning the temperature dependence of the oxygen non-stoichiometry was added to the revised manuscript. As presented in the updated Figure 3b, in all single-phase compositions, the level of oxygen content is nearly temperature-independent within the RT to 800 °C range. Such behavior can be considered typical for the near-stoichiometric, perovskite-related materials. Due to this, it is rather unlikely that the oxygen non-stoichiometry directly impacts the temperature dependence of the electrical conductivity, as no additional electron/hole charge carriers are needed to be introduced to the system as a part of charge compensating mechanisms. Instead, we would expect that the temperature dependence of the electrical conductivity will be impacted mainly by the band structure of the materials, as was described by J.B. Goodenough (Mat. Res. Bull. 8 (1973) 423-432). The discussion regarding this matter was added to the section concerning the results of Seebeck coefficient measurements.
Reviewer 2 Report
Reviewer: Presents a comprehensive work on the unique behavior that might provide new tools for design and properties’ tailoring of the Ruddlesden-Popper phases in the manuscript with entitled " Synthesis and properties of the gallium-containing Ruddles- 2 den-Popper oxides with high-entropy B-site arrangement ". The analysis done is detailed and well-organized, which gives plenty of its merits. However, the discussion is a bit poor, the authors should elaborate on the discussion of the results obtained. Additionally, the main problem with this article lies in the experimental design and statistical analysis.
Common comments are as follows:
1. The English writing needs to be improved and please double-check grammar and spelling.
2. The temperature dependence of the total conductivity is similar for all materials, with the highest conductivity value of 4.28 S/cm being reported for Sm-based composition. If possible, Incorporate piezoelectric data or magneto-responsive property.
3. You concluded “Based on the obtained results, it can be stated that the application of the high-entropy approach in these materials, resulted in prominent differences when compared with their more conventional counterparts, indicating a much more profound impact on the B-site occupancy, over the A-site one, than typically reported for these type of oxide systems”
Without the use of any statistical tools
3. The AFM analysis of the material surface must be performed.
Author Response
Dear Reviewer 2, thank you for your time and effort dedicated to our manuscript. Below, are presented our answers to the issues raised in the review.
Reviewer: Presents a comprehensive work on the unique behavior that might provide new tools for design and properties’ tailoring of the Ruddlesden-Popper phases in the manuscript with entitled " Synthesis and properties of the gallium-containing Ruddles- 2 den-Popper oxides with high-entropy B-site arrangement ". The analysis done is detailed and well-organized, which gives plenty of its merits. However, the discussion is a bit poor, the authors should elaborate on the discussion of the results obtained. Additionally, the main problem with this article lies in the experimental design and statistical analysis.
Both the introduction part and the discussion of the results were expanded, as suggested by the reviewer. Further information regarding the conventional, reference systems was added to the text, to provide a necessary context. The discussion regarding the oxygen non-stoichiometry and transport properties (especially the Seebeck coefficient) was rewritten, and new literature positions were added, to enable more detailed analysis.
Common comments are as follows:
1. The English writing needs to be improved and please double-check grammar and spelling.
Additional grammar and spelling check was performed.
2. The temperature dependence of the total conductivity is similar for all materials, with the highest conductivity value of 4.28 S/cm being reported for Sm-based composition. If possible, Incorporate piezoelectric data or magneto-responsive property.
While we agree with the Reviewer that such measurements would certainly provide additional, more detailed insight into the transport mechanism of the studied materials, unfortunately, at the moment we do not have an access to neither of the listed methods. Instead, certain aspects related to the transport properties of the Ruddlesden-Popper-type materials were added to the discussion in the section concerning the Seebeck coefficient value.
3. You concluded “Based on the obtained results, it can be stated that the application of the high-entropy approach in these materials, resulted in prominent differences when compared with their more conventional counterparts, indicating a much more profound impact on the B-site occupancy, over the A-site one, than typically reported for these type of oxide systems”
Without the use of any statistical tools
Unfortunately, this remark by the Reviewer appears to be incomplete in the system. Therefore, we assumed, that the Reviewer is concerned by the rather strong nature of the cited statement. Based on the aforementioned, expanded discussion of the results, we agree with the Reviewer that at this stage of studies, with no data available for other B-site disordered, high-entropy Ruddlesden-Popper oxides, it is impossible to unambiguously state that the reported behavior can be generalized on all materials of such type, as it might as well be distinct exclusively to the combination of B-site cations considered in the manuscript. Consequently, the conclusion part was revised to reflect this viewpoint:
“Based on the obtained results, it can be stated that the application of the high-entropy approach in these materials resulted in certain, distinct features when compared with their more conventional counterparts, indicating a more profound impact of the B-site occupancy over the A-site one, than typically reported for these type of oxide systems. However, at this stage of studies, it is impossible to state whether the reported features will be applicable to all high-entropy RP oxides, or if they are characteristic just to the studied series of materials. Nevertheless, the presented materials already appear to possess potentially beneficial characteristics from the viewpoint of high-temperature application, and what is even more important, the reported, unique behavior might provide new tools for design and properties’ tailoring of the Ruddlesden-Popper phases.”
To add to that, throughout the whole manuscript, additional information regarding the properties of conventional, RP-type oxides was added, to provide the necessary context for comparison between them and their high-entropy counterparts.
4. The AFM analysis of the material surface must be performed.
The AFM measurements were performed for all single-phase compositions, and the results were added to the Supplementary Information file (Figure S6).
Round 2
Reviewer 2 Report
Improved as per suggestion